# The Variant p.Ala84Pro Is Causative of X-Linked Hypophosphatemic Rickets: Possible Relationship with Burosumab Swinging Response in Adults

**DOI:** 10.3390/genes14010080

**Published:** 2022-12-27

**Authors:** Maria Carmela Zagari, Paola Chiarello, Stefano Iuliano, Lucia D’Antona, Valentina Rocca, Emma Colao, Nicola Perrotti, Francesca Greco, Rodolfo Iuliano, Antonio Aversa

**Affiliations:** 1Endocrinology Rare Disease Unit, Department of Experimental and Clinical Medicine, Magna Græcia University, Catanzaro, 88100 Catanzaro, Italy; 2Department of Pediatrics, Pugliese-Ciaccio Hospital, Catanzaro, 88100 Catanzaro, Italy; 3Department of Experimental and Clinical Medicine, Magna Græcia University of Catanzaro, 88100 Catanzaro, Italy; 4Department of Health Sciences, Magna Græcia University of Catanzaro, 88100 Catanzaro, Italy; 5Medical Genetics Unit, Mater Domini University Hospital, Catanzaro, 88100 Catanzaro, Italy; 6Department of Human Movement Sciences and Health, University of Rome “Foro Italico”, 00135 Rome, Italy

**Keywords:** burosumab, *PHEX* gene, XLH, rickets, FGF-23, quality of life

## Abstract

Loss of function mutations in the *PHEX* gene could determine X-linked dominant hypophosphatemia. This is the most common form of genetic rickets. It is characterized by renal phosphate wasting determining an increase in fibroblast growth factor 23 (FGF-23), growth retard, bone deformities and musculoskeletal manifestations. In recent decades, analysis of the *PHEX* gene has revealed numerous different mutations. However, no clear genotype-phenotype correlations have been reported in patients with hypophosphatemic rickets (XLH). We report two cases of a 28-year-old-male (patient 1) and a 19-year-old male (patient 2) affected by XLH initially treated with phosphate and 1,25-dihydroxyvitamin–D admitted to the Endocrinology unit because of the persistence of muscle weakness, bone pain and fatigue. After phosphate withdrawal, both patients started therapy with burosumab and symptoms ameliorated in three months. However, patient 1’s biochemical parameters did not improve as expected so we decided to investigate his genetic asset. We herein describe a possible clinical implication for the missense “de novo” mutation, c.250G>C (p.Ala84Pro) in the *PHEX* gene, reported in the *PHEX* database and classified as a variant of uncertain significance (VUS). The clinical implication of this mutation on disease burden and quality of life in adults is still under investigation.

## 1. Introduction

X-linked dominant hypophosphatemic rickets (XLH) is an uncommon genetic condition with an incidence of 1:20.000 individuals [1]. It is caused by loss-of-function mutations in the *PHEX* gene (phosphate-regulating endopeptidase homologue, X-linked) located on chromosome locus Xp2. Presently, more than 500 mutations of the *PHEX* gene have been identified [2]. The hallmark of the disease is represented by increased levels of FGF-23 and hypophosphatemia due to impaired phosphate conservation and excessive phosphate excretion [3]. Additionally, laboratory tests show elevated alkaline phosphatase [ALP in children, bone alkaline phosphatase (BAP) in adults], normal calcemia and lower than normal level of 1,25-dihydroxyvitamin D (1,25(OH)2D) [1]. This leads to skeletal abnormalities consistent with rickets, dental problems such as abscesses without caries and hearing loss [1,4]. The symptoms of XLH usually start in the first two years of life: rickets is a hallmark of XLH in children [3]. Adults with XLH can have symptoms such as enthesopathy and osteomalacia with increased risk of stress fractures and other complications, muscle weakness, pain and fatigue [1,3].

After growth, conventional therapy aims at improving symptoms and reducing the extent of osteomalacia [5]. It consists of active vitamin D analogs supplementation and oral phosphate with multiple daily intakes that do not specifically treat the underlying pathophysiology of elevated FGF-23-induced hypophosphatemia with limitations in management and effectiveness [5]. Indeed, it poorly corrects the biochemistry and symptoms of XLH, and has also been associated with different side effects such as gastrointestinal effects, secondary hyperparathyroidism, nephrocalcinosis, hypercalciuria and cardiovascular abnormalities [5]. Furthermore, orthopedic surgery to correct bone deformities is often needed [6]. 

Burosumab is a human monoclonal antibody (Ab) that binds FGF-23 and reduces its activity [7]. As stated before, burosumab has been approved since 2018 for the treatment of XLH children > 1 year of age and in adolescents with a growing skeleton by the European Medicines Agency (EMA) [8]. In April 2018, the US Food and Drug Administration (FDA) approved burosumab to treat also adult patients [9], based on the results of trials testing burosumab in children and adults with severe XLH and with bone pain and/or osteomalacia showing significantly increases serum phosphorus levels, biochemical markers of bone remodeling and improved histomorphometric indices of osteomalacia [3,9,10]. We report the cases of two young adult men with XLH. 

## 2. Patient and Methods

In May 2021 and April 2022, we enrolled two males, a 28-year-old and a 19-year-old, respectively, affected by XLH diagnosed after birth. Written informed consent for the publication of this case report was obtained from the patients. The study was performed in accordance with the Declaration of Helsinki. Blood samples were collected, and DNA was extracted with a commercial kit. All exons of *PHEX* were amplified by polymerase chain reaction (PCR) with specific primers [11] and sequenced with SeqStudio genetic analyzer as previously described [12]. All sequences were compared with NM_000444.6 as a reference sequence. All exons of FGF-23 gene were also sequenced with the same method and the sequences were then compared with NM_020638.3.

The six-minute walking test (6MWT) and handgrip strength test (HGS) were used to assess physical function and muscle strength, respectively [13,14]. The Western Ontario and McMaster Universities Arthritis Index (WOMAC) [15] was used to assess joint stiffness and functional limitations during daily activities, whereas the Brief Pain Inventory (BPI) was used to evaluate worst pain [16]. Higher scores indicate worse pain, stiffness, and functional limitations.

## 3. Results

### 3.1. Patient 1

Patient 1 is a 28-year-old male, referred to our Department of Endocrinology for XLH. His past medical history revealed failure to thrive and to achieve normal motor development since the first year of life, dolichocephaly, Harrison’s groove and progressive lower limb deformities. He was born by spontaneous birth at 40 weeks’ gestation. His birth weight was 3350 g (25th–50th percentile) and his birth height was 49 cm (10th–25th). Radiological findings displayed widening of the distal femoral metaphysis with genu-varum and poor definition of bone contours (Figure 1). Family history revealed no affected members across three generations. The first blood workup after birth showed hypophosphatemia (1.9 mg/dL), increased alkaline phosphatase (ALP) (1993 U/L) with normal 25-OH-vitamin D and 1,25-(OH)2D, calcium (9 mg/dL) and parathormone (PTH) (23 pg/mL) levels. Moreover, the tubular reabsorption of phosphate corrected for glomerular filtration rate (TmP/GFR) was reduced (1.254 mg/dL) for age-based normal reference interval (2.9–6.5 mg/dL).

At two years old, hypophosphatemic rickets was diagnosed so he started treatment with calcitriol 30 ng/Kg/die and elemental phosphorus 50 mg/Kg/die taken five times a day.

Conventional therapy was progressively adapted to the patient’s body weight as depicted in Figure 2. He followed a normal calcium intake (1.2 g per day) and a low-sodium diet. 

At three years old, he underwent surgery for decompression of the posterior cranial fossa due to prolapse of the cerebellar tonsils through the foramen magnum and underwent several surgical corrections of elongation of the lower legs with performing osteotomies using external fixation techniques.

At the time of our baseline examination—dose 0 (T_0_) he had a height of 160 cm and a body mass of 90 kg. HGS of the dominant upper limb was 38 kg_f,_ and the patient covered 293 m during the 6MWT. He reported a score of 3 out of 8 for joint stiffness and of 27 out of 68 for physical function limitation, whereas worst pain was severe according to BPI.

The laboratory tests at T_0_ showed hypophosphatemia (1.5 mg/dL), moderately elevated BAP (27.70 Ug/L) normal calcium level (9.6 mg/dL), elevated PTH (274.5 pg/mL) and low levels of 25-OH-vitamin D (7.3 ng/mL). Although the patient was still receiving conventional treatment with phosphates (1200 mg daily divided in three doses) and calcitriol (0.75 μg daily), there was only a slight improvement in osteomalacia, while gastrointestinal side effects, such as nausea, persisted. The patient still suffered skeletal and muscle pain which worsened his physical function and quality of life. Therefore, we discontinued phosphate and calcitriol treatment and decided to start therapy with burosumab in May 2021 at the maximum dose of 90 mg (of 1 mg/kg body weight), subcutaneously every 4 weeks. Moreover, we decided to combine his therapy with oral cholecalciferol 25,000 IU every 2 weeks (see Figure 2).

Burosumab reduced the worst bone pain (from severe to moderate pain) and improved walking distance during the 6MWT (from 293 to 314 m) and HGS (from 38 to 40 kg_f_) at week 24—dose 6 (T_1_) compared to T_0_. However, joint stiffness and physical function scores remained unchanged. Phosphatemia and TmP/GFR minimally increased at T_1_ from 1.5 mg/dL to 2.2 mg/dL and from 1.20 mg/dL to 1.94 mg/dL, respectively, C-terminal telopeptide (CTX) 1.89 ng/mL and BAP of 52.00 ug/L (n.v. 5.50–24.6) and normal 1,25-(OH)2 Vitamin D level (22.8 pg/mL). After six doses, we considered shortening the burosumab dosing interval to 3 weeks. After the seventh administration, the level of phosphate increased to 2.1 mg/dL and TmP/GFR reached 2.11 mg/dL (dose 8). The major biochemical findings are summarized in Table 1 and Table 2 and illustrated in Figure 3. The physical evaluation performed at dose 13 (T_2_) revealed an improvement in walking distance during the 6MWT (+31 m) and in HGS (+2 kg_f_) than at T_1_. Joint stiffness and physical function scores were lower at T_2_ than at T_1_(2 vs. 3 and 19 vs. 27 score, respectively) also revealing a reduction of pain (from moderate to mild pain). Physical and questionnaire results are reported in Table 3. 

Because biochemical parameters did not improve after 24 weeks (T_1_) from dose 1 of burosumab as expected, we decided to investigate patient 1’s genetic asset.

Gene analysis revealed a de novo *PHEX* gene mutation, registered in the *PHEX* database and reported in the ClinVar database as VUS. In the patient, the sequence of *PHEX* gene showed the presence of c.250G>C (p.Ala84Pro) variant, in a hemizygous state, positioned in the exon 3 of *PHEX* (see Figure 4). The variant is absent in the gnomAD [17] database and resulted as deleterious by the computational tool PredictSNP [18] with 87% of confidence. No variants of FGF-23 were found in the patient.

### 3.2. Patient 2

The patient is a 19-year-old boy who received a diagnosis of hypophosphatemic ricket at two years old. Gene analysis revealed the presence of c.118 + 1G > A mutation in exon 1 of *PHEX* gene. However, this mutation was not found in his parents. Immediately afterwards, he underwent cranial surgery because of scaphocephaly. 

At the time of admission, he had a height of 155 cm and a body mass of 79.5 Kg (BMI = 33.09 kg/cm^2^). He showed genu varum but no dental defects. He also performed limbs X-rays with evidence of multiple femur, radius and ulna deformities. 

At the age of 2 years old he started therapy with α-calcidiol (20 ng/kg/die) and inorganic phosphorus (30 mg/kg/die divided in four administrations). During his growth, therapy was progressively titrated, and until the time of admittance to our Unit, at 18 years old (see Figure 5). Prior therapy poorly corrected his bone defects and, moreover, it was associated with multiple side effects such as nausea and headache so we decided to start therapy with burosumab in May 2022 at the dose of 90 mg (1 mg/kg) every four weeks (Figure 5).

First blood workup at T_0_ (dose 0) revealed hypophosphatemia with normal levels of ionized calcium and 1,25-(OH)-2-D. By contrast, TmP/GFR was reduced (1.89 mg/dL) according to its age-related range (2.6–3.8 mg/dL). Every four weeks a complete blood workup was performed. Phosphate levels and TmP/GFR progressively ameliorated. Considering his mild reduced vitamin D levels, 25,000 UI cholecalciferol was administered every two weeks (major biochemical findings are shown in Table 4). 

Bone pain and physical impairment greatly improved throughout the follow-up period. Indeed, the physical evaluation performed after 10 weeks at dose 5 (T_2_) revealed an overall improvement in walking distance (+61 m) and in HGS (+6 kg_f_) than at T_0_. Moreover, joint stiffness and physical function scores were lower at T_2_ than at T_0_, revealing also a reduction of pain (from severe to mild pain). All improvements were greater when compared to those obtained in patient 1.

Physical evaluations and questionnaire scores are summarized in Table 5. 

## 4. Discussion

In this study we report a novel variant on the *PHEX* gene in an adult patient with XLH (Patient 1). To further discuss the possible pathogenic role of p.Ala84Pro, we found that a variant in the same residue of PHEX (p.Ala84Asp) associated with XLRH had been detected [19]. To investigate the mechanism of inheritance of the variant, we sequenced exon 3 of *PHEX* in the DNA extracted from the sample of the patient’s mother and we did not detect the variant (Figure 4). Thus, the variant of *PHEX* found in the patient 1 is de novo, in agreement with the absence of pathologic phenotype in the patient’s mother. Therefore, the c.250G>C (p.Ala84Pro) variant is classified as a likely pathogenic variant according to the guidelines of the American College of Medical Genetics and Genomics (ACMG) [20] with PS2, PM2, PM5 and PP3 criteria.

The mechanism of pathogenicity of this variant needs to be clarified. Missense variants have different effects on PHEX protein activity. Mechanisms of functional inactivation by missense variants of *PHEX* include inefficient glycosylation leading to defect in protein trafficking, lack of secretion in the extracellular medium and reduced peptidase activity [21].

We reported a pathogenic variant in the *PHEX* gene that supports the extreme variability in clinical and phenotypic presentation [22]. Patient 1 showed no dental defects and poor improvement in biochemical parameters due to persistent hypophosphatemia, with normal 1,25-dihydroxyvitamin D, and BAP during burosumab therapy administered every 4 weeks or every 3 weeks thereafter. 

To evaluate the efficacy and safety of therapy, normalization of the TmP/GFR ratio with higher serum phosphate levels, an increase in 1,25-dihydroxyvitamin D and an improvement in physical abilities should be pursued [3]. The low increase in phosphatemia levels has been attributed to the condition of secondary hyperparathyroidism that was promptly treated with cholecalciferol, 25,000 IU orally administered every two weeks during the first 6 months. However, 25-hydroxyvitamin D levels remained persistently low until dose 6, so that secondary causes were excluded. Thus, we decided to switch to the administration of 4000 IU/day of cholecalciferol in November 2021, which led only to a slight and progressive increase in 25-hydroxyvitamin D and a decrease in PTH plasma levels (Figure 2). 

To screen for possible tertiary hyperparathyroidism, neck and abdomen ultrasound were performed which turned out to be negative, so that the presence of either neck masses or nephrocalcinosis were excluded. Additionally, 24 h urinary calcium excretion persisted in normal range throughout the entire follow-up period (see Table 1 and Table 2). Despite the persistence of low serum phosphate levels, the patient reported a reduction in self-reported pain, stiffness, and physical limitations together with an improvement in 6MWT and HGS at the end of the last dose (T_2_) compared to T_0_ (Table 3). Musculoskeletal symptoms are known to be an important cause of physical distress in adults with XLH resulting in a reduced quality of life due to mobility impairments, loss of muscle strength, pain and reduced physical function [23,24]. Therefore, we evaluated the effect of burosumab on these endpoints using self-reported measures (WOMAC Index and BPI) together with physical assessments (6MWT and HGS). Based on our results, we hypothesized that improvements on physical outcomes may also reflect better self-reported scores. Therefore, these parameters could also be employed to evaluate therapy impact on physical function and self-reported symptoms. 

Patient 1 started therapy while he was still suffering a pre-existing condition of hypovitaminosis D and hyperparathyroidism, which, despite early treatment, is still under evolution. A very recent meta-analysis [25] suggested that in case of severe deficiency of 25-OH-D and high levels of PTH, the response to burosumab could improve after six injections [26].

Similarly, patient 2 started therapy with burosumab after conventional therapy was carried out from two years until adulthood. He showed improvements in physical activity and self-reported evaluations that commenced after 10-weeks from burosumab. Specifically, a marked improvement in 6 MWT and HGS were observed, respectively, as well as a reduction in pain and physical limitations scores. These results may rely on the younger age of the patient.

Efficacy of burosumab has been tested [27] in two siblings carrying on a variant with a clear loss-of-function mechanism. However, the mechanism of missense variants as that found in our study is probably more complex and needs to be elucidated. As previously reported in a murine model overexpressing high molecular weight FGF-2 isoforms, long-term anti-FGF-23-Ab therapy could not completely rescue the bone mineralization defect because of the overexpression of matrix extracellular phosphoglycoprotein (MEPE) and a consequent reduced bone remodelling [28].

Moreover, also in tumor-induced osteomalacia, expression of FGF-23 is accompanied by MEPE production [29]. In the missense variant p.Ala720Ser of *PHEX* gene there is an impairment in protein expression at the cell surface [30]. In addition, a clear genotype-phenotype correlation has not been established and, thus, accumulation of data is necessary to define a possible resistance to burosumab treatment correlated with the type or position of *PHEX* pathogenic variants. Interestingly, pathogenic variants localized in the first 649 amino acids of *PHEX* resulted correlated with a more severe phenotype and higher FGF23 levels [31]. In patient 1, since we did not detect any variant in the FGF-23 gene, a mechanism of resistance based on an imperfect binding of burosumab on FGF-23 is unlikely. Indeed, the same poor biochemical improvement was not experienced by patient 2, so we can speculate that the p.Ala84Pro variant may be one of the factors contributing to the swinging biochemical response to burosumab.

We reported a new clinical advance regarding the p.Ala84Pro variant in the *PHEX* gene, in a patient with uncertain biochemical response to burosumab but who achieved significant clinical benefits and improvements in his quality of life. Therefore, according to our data, we hypothesized that genetic analysis could help to establish the correlation between *PHEX* gene mutations, the phenotype and possible failure to clinical response to burosumab, also depending on very high baseline PTH levels. Whether the mutation reported by patient 1 may be relevant for the clinical response remains to be investigated with future transfection experiments aimed at studying gene function and modulation of gene expression and its functional role. 

## Figures and Tables

**Figure 1 genes-14-00080-f001:**
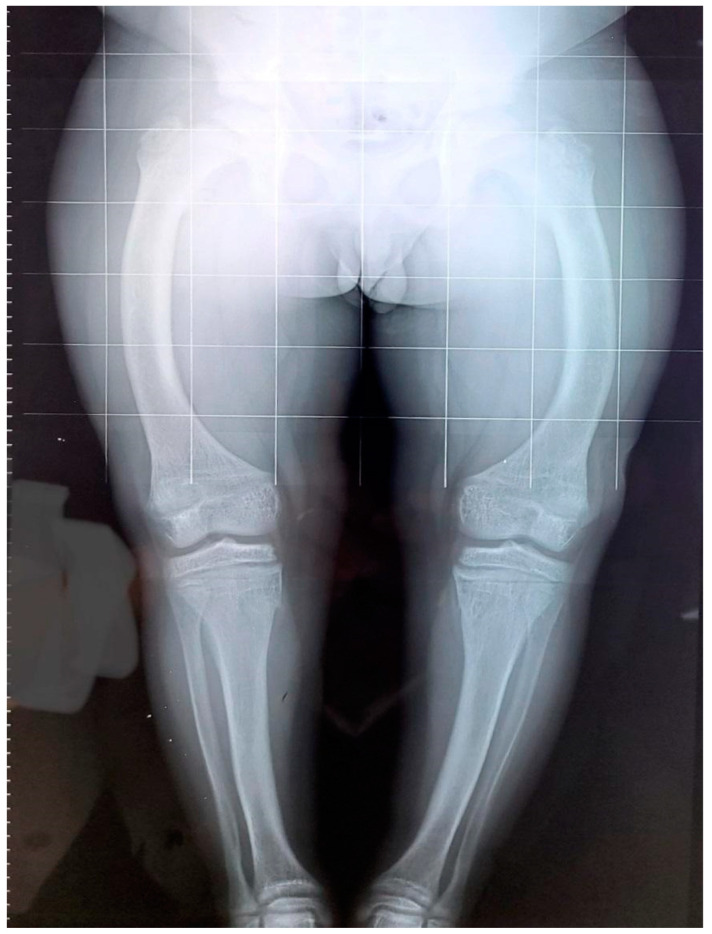
X-ray of the legs at 24 months of age with widening of the distal femoral metaphysis and genu varum.

**Figure 2 genes-14-00080-f002:**
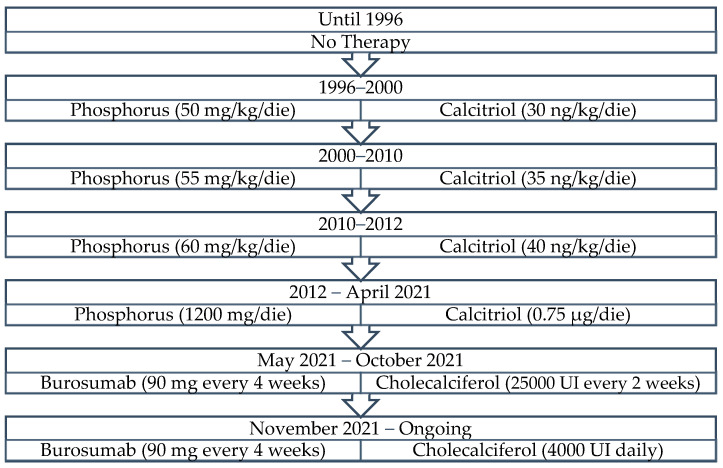
Patient 1—Graphic timeline of conventional therapy.

**Figure 3 genes-14-00080-f003:**
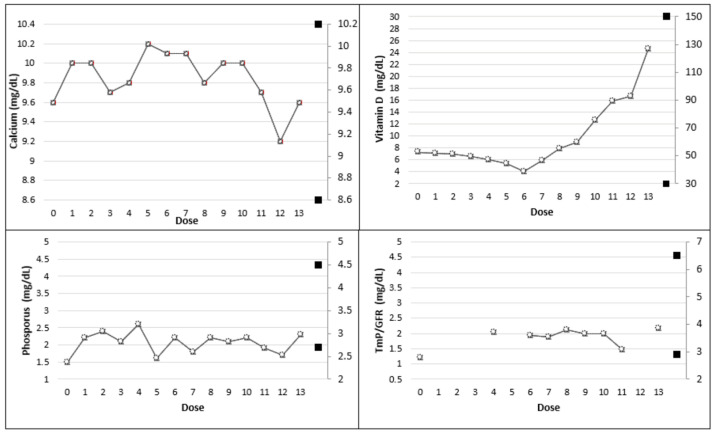
Trend of the main biochemical parameters (with age-related range in full bullets) in patient 1.

**Figure 4 genes-14-00080-f004:**
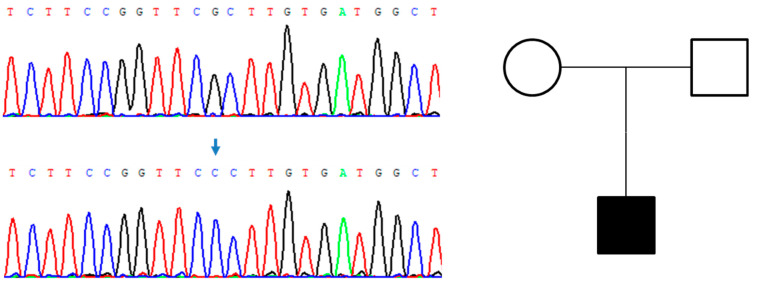
Genealogic tree of the patient’s family and electropherograms showing the region of exon 3 of *PHEX* containing the variant c.250G>C (p.Ala84Pro). The variant, indicated with an arrow, is present in the patient (bottom sequence) but not in his mother (top sequence).

**Figure 5 genes-14-00080-f005:**
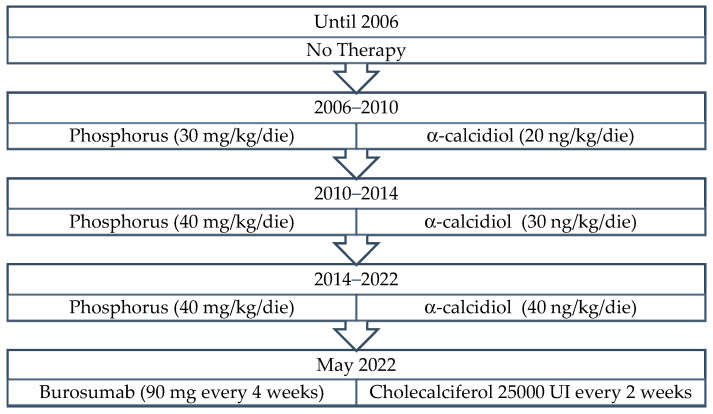
Patient 2—Graphic timeline of conventional therapy.

**Table 1 genes-14-00080-t001:** Patient 1—Biochemical findings over the follow up period until dose 6 (blood samples are collected the day before the next dose).

Dose							
	**Baseline—T_0_** **(No therapy)**	**1**	**2**	**3**	**4**	**5**	**6**
**Calcium** (8.6–10.2 mg/dL)	9.6	10.0	10.0	9.7	9.8	10.2	10.1
**Ionized Calcium** (4.7–5.2 mg/dL)	**4.8**	4.9	5.0	4.6	**4.7**	**5.1**	**5.0**
**Phosphorus** (2.7–4.5 mg/dL)	1.5	2.2	2.4	2.1	2.6	1.6	2.2
**Vitamin D** (30–150 ng/mL)	7.3	7.1	7.0	6.6	6.1	5.4	<4.0
**1,25-dihydroxyvitamin D** (19.9–79.3 pg/dL)	32.1	31.4	28.6	25.3	24.2	23.1	22.8
**PTH** (14–72 pg/mL)	274.5	245	239.7	189	148.4	274	276
**BAP** (5.50–24.6 ug/L)	27.7	-	-	-	47.5	-	-
**CTX** (<0.58 ng/mL)	2.91	2.42	1.92	1.82	1.89	1.57	1.44
**TmP/GFR** (2.9–6.5 mg/dL)	1.20	-	-	-	2.04	-	1.94
**Urinary Calcium (24 h)** (100–321 mg/24 h)	37	98	110	127	115	100	78

**Table 2 genes-14-00080-t002:** Patient 1—Biochemical findings over the follow up period, from dose 7 to dose 13 (blood samples are collected the day before the next dose).

Dose							
	**7**	**8**	**9**	**10**	**11**	**12**	**13**
**Calcium** (8.6–10.2 mg/dL)	10.1	9.8	10	10	9.7	9.2	9.6
**Ionized Calcium** (4.7–5.2 mg/dL)	5.0	**4.9**	5.0	5.0	4.9	4.6	4.8
**Phosphorus** (2.7–4.5 mg/dL)	1.8	2.2	2.1	2.2	1.9	1.7	2.3
**Vitamin D** (30–150 mg/dL)	5.9	7.9	9	12.7	15.9	16.6	24.6
**1,25-dihydroxyvitamin D** (19.9–79.3 pg/dL)	54.1	48.8	52.9	54.3	48.2	44.1	38.6
**PTH** (14–72 pg/mL)	152.2	142.4	191.2	174.5	142.8	181.6	188
**BAP** (5.50–24.6 ug/L)	52	53	52.4	57.4	-	69.4	-
**CTX** (<0.58 ng/mL)	1.42	1.69	1.89	1.88	1.82	2.11	2.13
**TmP/GFR** (2.9–6.5 mg/dL)	1.88	2.11	1.98	1.99	1.48	-	2.18
**Urinary Calcium (24 h)** (100–321 mg/24 h)	130	139	142	127	125	128	123

**Table 3 genes-14-00080-t003:** Patient 1—Physical evaluations and questionnaire scores.

Variable	T_0_	T_1_	T_2_
6MWT (m)	293	314(+7) *	345(+18) *
HGS-R (kg_f_)	38	40 (+5) *	42 (+11) *
Stiffness (score)	3/8	3/8 (0) *	2/8 (−33) *
Physical function (score)	27/68	27/68 (0) *	19/68 (−30) *
Worst pain	Severe	Moderate	Mild

T_0_, at baseline; T_1_, after 24 weeks (doses 6); T_2_, after 45 weeks (dose 13); R, right; 6MWT, six minute walking test; HGS, handgrip strength; * %difference vs. T_0_.

**Table 4 genes-14-00080-t004:** Patient 2—Biochemical findings over the follow-up period.

Dose							
	**Baseline—T_0_** **(No therapy)**	**1**	**2**	**3**	**4**	**5**	**6**
**Calcium** (8.6–10.2 mg/dL)	9.3	9.6	9.5	9.5	9.2	9.1	9.3
**Ionized calcium** (4.7–5.2 mg/dL)	4.6	4.7	4.8	4.8	-	4.9	4.6
**Phosphorus** (2.7–4.5 mg/dL)	1.6	2.4	2.8	2.82	2.8	2.6	2.6
**Vitamin D** (30–150 ng/mL)	19.9	17.1	19.3	21.3	25.6	33.5	19.9
**1,25-dihydroxyvitamin D** (19.9–79.3 pg/dL)	27.1	66.8	-	60.2	54.3	-	18.1
**PTH** (14–72 pg/mL)	109.7	128.9	121.8	105	103.2	100.5	109.7
**BAP** (5.50–24.6 ug/L)	67.7	52.2	54.8	49.6	48.8	-	67.7
**CTX** (<0.58 ng/mL)	1.95	4.36	4.31	3.43	2.94	1.84	-
**TmP/GFR** (2.9–6.5 mg/dL)	1.89	2.4	-	2.8	2.7	2.6	2.8
**Urinary calcium (24 h)** (100–321 mg/24 h)	115	195	102	127	-	137	131

**Table 5 genes-14-00080-t005:** Patient 2—Physical evaluations and questionnaire scores.

Variable	T_0_	T_1_	T_2_
6 MWT (m)	404	439 (+9) *	465 (+15) *
HGS-R (kg_f_)	40	43 (+8) *	46 (+15) *
Stiffness (score)	2/8	2/8 (0%) *	1/8 (+50) *
Physical function (score)	24/68	20/68 (−17) *	15/68 (−38) *
Worst pain	Severe	Moderate	Mild

T_0_, at baseline; T_1_, at dose 3 (6 weeks); T_2_, at dose 5 (10 weeks); R, right; 6MWT, six minute walking test; HGS, handgrip strength; * %difference vs. T_0._

## Data Availability

Data supporting the reported results are available from the corresponding author upon reasonable request.

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
