# Peer review of "The Variant p.Ala84Pro Is Causative of X-Linked Hypophosphatemic Rickets: Possible Relationship with Burosumab Swinging Response in Adults"

_genes, 2022, doi:10.3390/genes14010080_

Round 1
Reviewer 1 Report
The Authors have described an interesting case report regarding a new variant in the PHEX gene (p.Ala84Pro) in XLH and they suppose eventually a possible relationship with Burosumab to explain its swing response.
The strength of this paper is the genetic evaluation, which can discover more information regarding this master gene.
However, the clinical details you provide obscure and dilute this important aspect. In addition, some management you have reported is inappropriate.
So, please confine yourself to one subject.
The following issues should be revised before considering this paper suitable for a definitive publication:
-normal calcium level (10 mg/dL), elevated PTH (274,5 pg/ml), have you evaluated the ionized calcium?
As an aspect, the long-term use of calcitriol can drive an increase in PTH, until tertiary hyperparathyroidism forms. The slight increase of calcium (ionized calcium should be more appropriate to show) with an increase in PTH suggests this evaluation;
-“secondary hyperparathyroidism requires ultrasound”, please remove this sentence, because that is not evidence-based medicine and it generates confusion.
While the evaluation for nephrolithiasis/nephrocalcinosis is rationale, there is no mention of calcium urine 24 h excretion.
After the use of Burosumab, an expected increase in phosphate level was achieved due to the improvement of TmP/GFR, similar to show by clinical trial.
About increase PTH, the Authors have to keep in mind that in the clinical trial, plasma PTH levels decreased in the Burosumab group only at week 24! In the text is not clear how long the patient had been receiving the Burosumab, but considering the usual clinical management, the sentence “After the second administration……., showing no evident biochemical efficacy” must be remove.
Are you sure that evaluating FGF 23 during Ab FGF 23 treatment makes sense?
Indeed, some Authors have shown altered results due to possible immunological interference in immunoassays (Clin Chem Lab Med 2020; 58(11): e267–e269; https://doi.org/10.1515/cclm-2020-0460).
Regarding pathogenic PHEX mutation in your sporadic case of XLH, no wonder, that in another novel germline PHEX mutation (c.2158G>T; p.Ala720Ser) an impairment trafficking of the protein expression at the cell surface (compared for WT PHEX) was described (Bone 106 (2018) 30-34; doi.org/10.1016/j.bone.2017.10.002).
A few lines about this consideration should be added.
Have you considered if this mutation can also alter the regulation of other phosphatonins (e.g. MEPE) that are regulated also by PHEX? Indeed, in Tumor-induced osteomalacia, an acquired condition that from a physio-pathological point of view is analogous to XLH, the co-expression of MEPE has been reported (Ear Nose Throat J; 2022 Jun;101(5):289-291; DOI: 10.1177/0145561320940869).
Indeed, in a murine model, the long-term treatment with FGF23Ab further increased SIBLING protein-related genes and pyrophosphate-related genes in the bone that could contribute to the incomplete rescue of the mineralization defect (JBMR, Vol. 33, No. 7, July 2018, pp 1347–1361 DOI: 10.1002/jbmr.3417).
Finally, future log regarding the alternative management of your patient is not relevant, please remove it.
Author Response
Reply to reviewer 1
- Normal calcium level (10 mg/dL), elevated PTH (274,5 pg/ml), have you evaluated the ionized calcium?
According to this suggestion, we added ionized calcium throughout the manuscript and in the tables.
- As an aspect, the long-term use of calcitriol can drive an increase in PTH, until tertiary hyperparathyroidism forms. The slight increase of calcium (ionized calcium should be more appropriate to show) with an increase in PTH suggests this evaluation;
As suggested by the reviewer, to better clarify this aspect, we reformulated the sentence (page 9, lines 319, 320).
- “Secondary hyperparathyroidismrequires ultrasound”, please remove this sentence, because that is not evidence-based medicine and it generates confusion.
As stated before, we reformulated the sentence (page 9, line 319, 320).
- While the evaluation for nephrolithiasis- nephrocalcinosis is rationale, there isno mention of calcium urine 24 h excretion.
As suggested, we updated both the text and the tables (page 9, lines 321,322; tables 1, 2 and 4).
- In the text is not clear how long the patient had been receiving the Burosumab, but considering the usual clinical management, the sentence “After the second administration……., showing no evident biochemical efficacy” must be remove.
As proposed by reviewer the text has been reformulated and the sentence “After the second administration… showing no evident biochemical efficacy” has been removed (page 4, line 170). Moreover, we better clarified therapy patient 1 time interval in the text and in figure 2.
- Are you sure that evaluating FGF 23 during Ab FGF 23 treatment makes sense? Indeed, some Authors have shown altered results due to possible immunological interference in immunoassays (Clin Chem Lab Med 2020; 58(11): e267–e269; https://doi.org/10.1515/cclm-2020-0460).
Due to the lack of clear results, we removed FGF-23 dosage both from the text and from the tables.
- Regarding pathogenic PHEX mutation in your sporadic case of XLH, no wonder, that in another novel germline PHEX mutation (c.2158G>T; p.Ala720Ser) an impairment trafficking of the protein expression at the cell surface (compared for WT PHEX) was described (Bone 106 (2018) 30-34; doi.org/10.1016/j.bone.2017.10.002). A few lines about this consideration should be added.
We thank the reviewer for suggestion and so we updated the text accordingly (page 9, lines 479-481).
- Have you considered if this mutation can also alter the regulation of other phosphatonins (e.g. MEPE) that are regulated also by PHEX? Indeed, in Tumor-induced osteomalacia, an acquired condition that from a physio-pathological point of view is analogous to XLH, the co-expression of MEPE has been reported (Ear Nose Throat J; 2022 Jun;101(5):289-291; DOI: 1177/0145561320940869). Indeed, in a murine model, the long-term treatment with FGF23Ab further increased SIBLING protein-related genes and pyrophosphate-related genes in the bone that could contribute to the incomplete rescue of the mineralization defect (JBMR, Vol. 33, No. 7, July 2018, pp 1347–1361 DOI: 10.1002/jbmr.3417).
As suggested by the reviewer, we reported the possible relationship between MEPE and poor clinical improvement (page 9, lines 347-351).
- Finally, future log regarding the alternative management of your patient is not relevant, please remove it.
We agree with the reviewer’s opinion and have deleted the sentence regarding the future management of our patient.
Reviewer 2 Report
In the manuscript titled “A NEW VARIANT IN PHEX GENE p.Ala84Pro AND POSSIBLE RELATIONSHIP WITH BUROSUBAM SWINGING RESPONSE IN ADULT: A CASE REPORT”, the authors reported one adult XLH patient with a novel variant in the PHEX gene. They described his clinical course in detail, and the reviewer agrees that the accumulation of these data is important to clear the genotype-phenotype relationship not only in the natural history but also the response to burosumab. Unfortunately, however, this p.Ala84Pro variant has already been registered in the PHEX database as variant ID 93 (https://www.rarediseasegenes.com/variant/107)
Author Response
Reply to reviewer 2
In the manuscript titled “A NEW VARIANT IN PHEX GENE p.Ala84Pro AND POSSIBLE RELATIONSHIP WITH BUROSUMAB SWINGING RESPONSE IN ADULT: A CASE REPORT”, the authors reported one adult XLH patient with a novel variant in the PHEX gene. They described his clinical course in detail, and the reviewer agrees that the accumulation of these data is important to clear the genotype-phenotype relationship not only in the natural history but also the response to burosumab. Unfortunately, however, this p.Ala84Pro variant has already been registered in the PHEX database as variant ID 93 (https://www.rarediseasegenes.com/variant/107)
We thank the reviewer for the information provided. The PHEX variant p.Ala84Pro was registered in the PHEX database as VUS. In our case report, we provide new genetic evidence showing that this variant can be upgraded to Lykely pathogenic.
We modified the title in “The variant p.Ala84Pro is causative of X-linked hypophosphatemic rickets: possible relationship with burosumab swinging response in adult: a case report” and the text of the manuscript accordingly.
Reviewer 3 Report
The authors present a case report describing their clinical findings in a 28-year-old patient carrying a novel likely pathogenic “de novo” mutation of the PHEX gene. They initiated treatment with Burosumab, which led to reduced bone pain and improved physical functions, but did not fully correct the impaired biochemical parameters known to be present in X-linked hypophosphatemic rickets (the disease caused by PHEX inactivation). Although the data are overall well-presented and discussed in the context of the current knowledge, the novelty of the reported findings is limited. There are also some aspects of the manuscript, which should be improved.
Specific comments:
1) The introduction starts with the statement that X-linked dominant hypophosphatemic rickets is “mostly” caused by loss-of-function mutations in the PHEX gene. This implies that other forms of X-linked dominant hypophosphatemic rickets exist, which are not caused by PHEX inactivation. I don´t think that there is any evidence for this yet.
2) It is quite confusing to follow how the patient was treated before initiation of Burosumab therapy. There is information about the start of the treatment at the age of 2 years and about the slightly different treatment before Burosumab injections were started, i.e. at the age of 28 years. It is further stated that the patients followed, at baseline examination, a normal calcium intake and a low-sodium diet. The confusion is further enhanced by the 4th paragraph of the discussion, which does not relate to the statements in the Results section. I would strongly suggest to include a schematic presentation showing the timeline of treatment modifications. In particular, it needs to be clarified, if the patient was treated with calcitriol and elemental phosphorus for 26 years, before being examined by the authors.
3) Although it is respectable that the authors provide all values of their biochemical analyses during the time of Burosumab treatment in Table 1.2, it might be worthwhile to illustrate these data in form of line graphs (including highlighted reference ranges).
4) It is also unclear if the genetic analysis was performed before Burosumab treatment was initiated or thereafter (as it appears from the way of presentation in the Results section). This should be clarified.
5) Since PHEX sequencing was only performed for the patient and his mother, there is essentially only suggestive evidence (since the mutation is classified as likely pathogenic) for X-linked hypophosphatemic rickets. In this regard it is surely appropriate that the authors also sequenced all exons of the FGF23 gene, yet they should do the same for other genes mutated in hypophosphatemic disorders.
Author Response
Reply to reviewer 3
- The introduction starts with the statement that X-linked dominant hypophosphatemic rickets is “mostly” caused by loss-of-function mutations in the PHEX This implies that other forms of X-linked dominant hypophosphatemic rickets exist, which are not caused by PHEX inactivation. I don´t think that there is any evidence for this yet.
As proposed by reviewer 3, we removed “mostly” from the text (page 1, line 39).
- It is quite confusing to follow how the patient was treated before initiation of Burosumab therapy. There is information about the start of the treatment at the age of 2 years and about the slightly different treatment before Burosumab injections were started, i.e. at the age of 28 years. It is further stated that the patients followed, at baseline examination, a normal calcium intake and a low-sodium diet. The confusion is further enhanced by the 4thparagraph of the discussion, which does not relate to the statements in the Results section. I would strongly suggest to include a schematic presentation showing the timeline of treatment modifications. In particular, it needs to be clarified, if the patient was treated with calcitriol and elemental phosphorus for 26 years, before being examined by the authors.
As proposed by the reviewer, we updated the text to better clarify this aspect (page 5, lines 121-179) and we added figure 2.
- Although it is respectable that the authors provide all values of their biochemical analyses during the time of Burosumab treatment in Table 1.2, it might be worthwhile to illustrate these data in form of line graphs (including highlighted reference ranges).
As suggested, we inserted a new figure to better define variations occurred to major biochemical parameters across the 13 doses (see figure 3).
- It is also unclear if the genetic analysis was performed before Burosumab treatment was initiated or thereafter (as it appears from the way of presentation in the Results section). This should be clarified.
As required, we specified that genetic analysis has been performed 24 weeks after the initiation of Burosumab therapy (page 4, lines 178, 179).
- Since PHEXsequencing was only performed for the patient and his mother, there is essentially only suggestive evidence (since the mutation is classified as likely pathogenic) for X-linked hypophosphatemic rickets. In this regard it is surely appropriate that the authors also sequenced all exons of the FGF23 gene, yet they should do the same for other genes mutated in hypophosphatemic disorders.
We thank the reviewer for this important comment. Mutations in PHEX gene are the major cause of hypophosphatemic rickets (Nat Rev Nephrol. 2019 Jul;15(7):435-455). Since we found a variant that fulfils the criteria of pathogenicity, we did not sequence other minor causative genes. However, we sequenced FGF23 gene and we did not find any mutation, as reported also in the previous version of the manuscript (page 2, line 82).
Round 2
Reviewer 1 Report
The manuscript has been improved by Authors, so now it can be suitable for publication.
Reviewer 2 Report
Comments to authors,
In the revised manuscript titled “THE VARIANT p.Ala84Pro IS CAUSATIVE OF X-LINKED HYPOPHOSPHATEMIC RICKETS: POSSIBLE RELATIONSHIP WITH BUROSUBAM SWINGING RESPONSE IN ADULT: A CASE REPORT”, the authors modified their title and added another case with XLH. The added case has c.118+1G>A variant in exon 1 of PHEX gene. Because the variant of case 2 was different from p.Ala84Pro, the authors cannot explain that the p.Ala84Pro is causative of XLH by adding case 2. Unfortunately, the authors did not resolve their problem that the sample number was too small (only one) to publish their manuscript in this paper.
Reviewer 3 Report
The authors have adequately responded to my comments and signficantly improved their mansucript.